Reef manta rays forage on tidally driven, high density zooplankton patches in Hanifaru Bay, Maldives

Armstrong Asia O. asia.armstrong@uqconnect.edu.au 1
Stevens Guy M.W. 2
Townsend Kathy A. 3
Murray Annie 2
Bennett Michael B. 1
Armstrong Amelia J. 1
Uribe-Palomino Julian 4
Hosegood Phil 5
Dudgeon Christine L. 1 3
Richardson Anthony J. 4 6
1 School of Biomedical Sciences, The University of Queensland , St Lucia , Queensland , Australia
2 The Manta Trust , Catemwood House, Norwood Lane, Corscombe , Dorset , United Kingdom
3 School of Science, Technology, and Engineering, University of Sunshine Coast , Hervey Bay , Queensland , Australia
4 Queensland Biosciences Precinct, CSIRO Oceans and Atmosphere , St Lucia , Queensland , Australia
5 School of Biological and Marine Sciences, University of Plymouth , Plymouth , Devon , United Kingdom
6 School of Mathematics and Physics, The University of Queensland , St Lucia , Queensland , Australia
Hedrick Ann
Electronic publication date: 2021 Aug 23
Publication date: 2021
Volume: 9
Electronic Location ID: e11992
Received 2021 Apr 28; Accepted 2021 Jul 27
Copyright: ©2021 Armstrong et al.
Copyright year: 2021
Copyright holder: Armstrong et al.
License: This is an open access article distributed under the terms of the Creative Commons Attribution License, which permits unrestricted use, distribution, reproduction and adaptation in any medium and for any purpose provided that it is properly attributed. For attribution, the original author(s), title, publication source (PeerJ) and either DOI or URL of the article must be cited.
License URL: https://creativecommons.org/licenses/by/4.0/

Keywords: Foraging ecology, Foraging threshold, Megafauna, Zooplanktivore, Mobulid ray, Undinula, ZooScan

Funding: Australian Research Council LP150100669 University of Queensland Research Scholarships Four Seasons Resorts Maldives at Landaa Giraavaru This study was funded by the Australian Research Council, Grant/Award Number: LP150100669. Asia O. Armstrong and Amelia J. Armstrong were funded by University of Queensland Research Scholarships. This study was made possible due to funding from Carl F. Bucherer, the Save Our Seas Foundation, and the logistical support and funding from the Four Seasons Resorts Maldives at Landaa Giraavaru. The funders had no role in study design, data collection and analysis, decision to publish, or preparation of the manuscript.

==============================
Manta rays forage for zooplankton in tropical and subtropical marine environments, which are generally nutrient-poor. Feeding often occurs at predictable locations where these large, mobile cartilaginous fishes congregate to exploit ephemeral productivity hotspots. Investigating the zooplankton dynamics that lead to such feeding aggregations remains a key question for understanding their movement ecology. The aim of this study is to investigate the feeding environment at the largest known aggregation for reef manta rays Mobula alfredi in the world. We sampled zooplankton throughout the tidal cycle, and recorded M. alfredi activity and behaviour, alongside environmental variables at Hanifaru Bay, Maldives. We constructed generalised linear models to investigate possible relationships between zooplankton dynamics, environmental parameters, and how they influenced M. alfredi abundance, behaviour, and foraging strategies. Zooplankton biomass changed rapidly throughout the tidal cycle, and M. alfredi feeding events were significantly related to high zooplankton biomass. Mobula alfredi switched from non-feeding to feeding behaviour at a prey density threshold of 53.7 mg dry mass m−3; more than double the calculated density estimates needed to theoretically meet their metabolic requirements. The highest numbers of M. alfredi observed in Hanifaru Bay corresponded to when they were engaged in feeding behaviour. The community composition of zooplankton was different when M. alfredi was feeding (dominated by copepods and crustaceans) compared to when present but not feeding (more gelatinous species present than in feeding samples). The dominant zooplankton species recorded was Undinula vulgaris. This is a large-bodied calanoid copepod species that blooms in oceanic waters, suggesting offshore influences at the site. Here, we have characterised aspects of the feeding environment for M. alfredi in Hanifaru Bay and identified some of the conditions that may result in large aggregations of this threatened planktivore, and this information can help inform management of this economically important marine protected area.

Introduction

Manta rays are large planktivores that inhabit tropical and subtropical waters globally, which are generally oligotrophic (Marshall, Compagno & Bennett, 2009). Therefore, to meet their metabolic needs, manta rays need to locate pulses of zooplankton productivity. Similar to other tropical planktivores, such as leatherback turtles Dermochelys coriacea (Hays et al., 2006) and whale sharks Rhincodon typus (Rohner et al., 2015), manta rays aggregate where and when conditions result in elevated local productivity (Dewar et al., 2008; Anderson, Adam & Goes, 2011; Jaine et al., 2012). However, these productivity ‘hotspots’ are ephemeral in nature and often difficult for researchers to locate and characterise (Harris et al., 2020; Harris et al., 2021), which makes the direct study of planktivore feeding ecology challenging (Sims, 2008; Rohner et al., 2015).

A variety of approaches are used to study a species’ diet, including stomach contents analysis, biochemical analyses, and direct observation. Two studies have recently explored the diet of manta ray species based on stomach contents: one on oceanic manta rays Mobula birostris taken in a fishery in the Philippines (Rohner et al., 2017), and one on a historic stomach sample from a reef manta ray M. alfredi collected from eastern Australia (Bennett et al., 2017). Traditionally, lethal approaches for dietary analysis, such as stomach contents analysis, are inappropriate for vulnerable marine fishes (Cortés, 1997), and only offer a ‘snapshot’ of a species’ diet (Rohner et al., 2013). Instead, biochemical approaches, including stable isotope and fatty acid analysis, are non-lethal methods that provide an integrated signal that represents the long-term diet and trophic position of species. Biochemical analysis has inferred that M. birostris off Ecuador derive much of its calorific intake by feeding at depth (Burgess et al., 2016), as does M. alfredi off eastern Australia (Couturier et al., 2013), and M. alfredi in the Seychelles targets pelagic zooplankton sources (Peel et al., 2019). Nevertheless, biochemical analyses lack resolution, such as identifying and quantifying preferred prey species, whereas direct observation of animal feeding allows simultaneous sampling of the feeding environment (Sims & Merrett, 1997; Rohner et al., 2015; Fortune et al., 2020).

Currently, the only detailed direct observation of the diet in manta rays is from an aggregation site off eastern Australia, where M. alfredi was observed feeding near the surface (Armstrong et al., 2016). The study found M. alfredi feeding events were significantly associated with greater zooplankton biomass, but were not influenced by zooplankton size or species composition. Further, feeding activity and zooplankton density was tidally driven at this site. Similarly, the occurrence of M. alfredi in Komodo National Park in Indonesia was heavily influenced by tide, and was considered likely to be related to feeding activity (Dewar et al., 2008). An in situ prey density threshold of 11.2 mg m−3 was determined for M. alfredi foraging in eastern Australia (Armstrong et al., 2016). However, a theoretical estimate of the density threshold to meet their metabolic requirements (25.2 mg m−3) suggests they require additional energy from alternate food sources, such as foraging at depth (Armstrong et al., 2016).

Manta rays exhibit behavioural plasticity in relation to their feeding environment. In eastern Australia (Jaine et al., 2012), Indonesia (Dewar et al., 2008), and the Chagos Archipelago (Harris et al., 2021), surface feeding by manta rays is frequently observed during daylight hours, and other large planktivores, such as basking sharks Cetorhinus maximus (Sims & Merrett, 1997) and R. typus (Prebble, 2018) also employ this strategy. In the Red Sea, M. alfredi swim in various circular patterns when feeding on zooplankton in shallow water (Gadig & Neto, 2014). At Ningaloo Reef in Western Australia, M. alfredi is frequently observed to use a combination of feeding modes, including surface feeding, somersaulting, and bottom feeding (AOA pers. obs.). A unique strategy of “cyclone” feeding has been described at Hanifaru Bay in the Maldives, where multiple individuals manipulate the water column to create a vortex that concentrates zooplankton on which they then feed (Stevens, 2016). Eight different feeding strategies have been described at this site, and have been related to prey density using a subjective visual assessment of the water column (Stevens et al., 2018). However, zooplankton density or composition has yet to be quantified in relation to these strategies. Upwards of 250 individual manta rays aggregate in Hanifaru Bay during peak feeding events, making it the largest known M. alfredi aggregation site in the world (Harris et al., 2020). This is therefore an ideal location to test hypotheses regarding habitat use, aggregative behaviour, feeding strategies, and zooplankton dynamics for this species.

Here, we investigate the food environment of M. alfredi at Hanifaru Bay, an ecologically and economically important marine protected area and core zone within a UNESCO Biosphere Reserve. Manta rays are of economic importance to both ecotourism and fisheries industries and have a conservative life history, and so identifying the foraging requirements and habitat preferences of these threatened rays should aid future conservation efforts (Stewart et al., 2018). We analyse the zooplankton dynamics (biomass, size structure, and community composition) in relation to the presence, behaviour and feeding strategies of M. alfredi, to improve our understanding of the feeding dynamics of this large planktivorous species. This study aims to relate changes in zooplankton biomass to M. alfredi behaviour; to establish a critical prey density threshold for feeding at this site, and to determine whether prey density influences the type of feeding strategy M. alfredi employs to exploit their prey. Further, we aim to investigate whether M. alfredi foraging behaviour is influenced by changes in the zooplankton community composition, or size structure.

Materials & Methods

Study site

The Maldives has a large resident population of M. alfredi which undertake biannual migrations linked to the changing monsoons (seasons) within the archipelago (Anderson, Adam & Goes, 2011; Fig. 1A). During the Southwest Monsoon, or Hulhangu (April–November), numerous M. alfredi frequent foraging aggregation sites on the eastern side of the nation’s atolls (Harris et al., 2020). One site, Hanifaru Bay, is situated on the eastern edge of Baa Atoll, and attracts large feeding aggregations of this species annually (Stevens, 2016; Harris et al., 2020). Hanifaru Bay is a small reef inlet (700 m long by 200 m wide) which forms part of a core marine protected area within the Baa Atoll UNESCO Biosphere Reserve (5°17′N, 73°15′E; Fig. 1B). The shallow (maximum depth 22 m) inlet is periodically inundated with zooplankton-rich water. Motorised boat activity and SCUBA diving are prohibited in Hanifaru Bay due to the high numbers of manta rays and other megafauna that access the inlet (Murray et al., 2020).

Figure 1 Study site in Hanifaru Bay in Baa Atoll, The Maldives.

(A) Map of The Maldives, black star indicates location of Hanifaru Bay in Baa Atoll; and (B) Satellite image of Hanifaru Island and Lagoon, with key study site of Hanifaru Bay and prevailing current regimes (Credit: Copernicus Sentinel data 2020, accessed via USGS EarthExplorer and processed by Amelia J Armstrong).

Data collection

Fieldwork was conducted in the lead up to the new Moon in August 2017, under Ministry of Fisheries Permit No. (OTHR)30-D/PRIV/2017/280, and Ministry of Environment Permit No’s. EPA/2017/RP-01 & EPA/2016/PSR-M02. This time of year was chosen because strong lunar tides appear to overcome the force of the prevailing monsoonal current, drawing plankton-rich water from outside the atoll edge into Hanifaru Bay (Harris & Stevens, 2021). The currents form a back eddy, trapping and concentrating plankton in this shallow reef inlet, resulting in M. alfredi foraging opportunities, which peak during spring and high tides (Stevens, 2016; Harris & Stevens, 2021). Sampling was conducted during daylight hours and across the tidal cycle from 13–21 August 2017. Zooplankton was collected by two people using a 200 µm-mesh net with a 50 cm diameter mouth. The net was towed by hand at the surface for a ∼50 m transect between two coral features at the eastern end of Hanifaru Bay (Fig. 2A), at a speed of ∼22 m per minute. A flowmeter was fitted to the plankton net to allow calculation of the volume of water sampled. Flowmeter calibration was performed prior to the field trip in a swimming pool of known length to establish an accurate measurement of distance per flowmeter revolution. Samples were kept on ice and fixed with 10% buffered formalin solution at the end of each day.

Figure 2 Zooplankton sampling and reef manta ray Mobula alfredi feeding strategies.

(A) Zooplankton samples were collected by two snorkellers surface swimming a 200 µm-mesh net with flowmeter for 50 m; and (B) Ethogram of feeding strategies: (i) Straight, (ii) Surface, (iii) Chain, (iv) Piggy-back, (v) Somersault, (vi) Cyclone, (vii) Sideways, and (viii) Bottom (Illustration credit: Marc Dando).

Each net tow was accompanied by an in-water observer recording manta ray activity in vicinity of the tow. This included: (1) manta ray abundance; (2) behaviour (Feeding, Non-feeding—when manta rays were present but not feeding, and Absent—when manta rays were not present); and (3) most common feeding strategy employed (as described in (Stevens, 2016; Fig. 2B).

Temperature and salinity data were collected at 1 s intervals from a CTD unit deployed at the site for the study duration (except for ∼24 hrs from 17–18 August for battery exchange). Temperature and salinity ranges were relatively small throughout the study (28.6–29.2 °C and 34.3–35.0 ppt respectively). These variables were excluded from the models as their inclusion resulted in missing values. Local tide data were obtained from a government representative from the Ministry of Environment.

Sample processing

Zooplankton samples were processed in the CSIRO Plankton Ecology Laboratory in Brisbane, Australia. Flowmeter readings and the area of the net mouth were used to estimate the volume of water filtered.

Zooplankton biomass

Zooplankton samples were split into two halves using a Folsom splitter (Harris et al, 2000). The first half was used to determine dry mass, with each sample oven-dried at 70 °C for 24 hrs prior to weighing. Zooplankton dry mass (hereafter referred to as biomass) per unit volume of filter-seawater for each tow was calculated by dividing the dry mass of the sample (mg) by the volume of filtered water (m3): Biomassmgm−3=Drymassmg/Volumeofwaterfilteredm3.

Zooplankton identification

The second half of the sample was used to examine size structure and community composition via a 2400 dpi ZooScan system and microscopy. The Hydroptic v3 ZooScan (EPSON Perfection V700 Flatbed) is a high resolution, waterproof scanner that digitises particles for size and biovolume measurements (Gorsky et al., 2010). An aliquot of each sample was prepared using a Stemple pipette of known volume and placed on the scanning tray. To avoid overlap, particles were manually separated. Once separated, the sample was scanned and particles were extracted into vignettes for categorisation into broad taxonomic groups (24 groups) using Plankton ID software (Version 1.2.6) and manual validation (Gorsky et al., 2010). Objects classified as sand, fibre, detritus, bubbles and shadows were excluded from further analysis (as per (Rohner et al., 2015). For visualisation, taxa that comprised <5% of the total abundance were grouped as “other”, and these included cnidaria, polychaetes, echinoderm larvae, bryozoan larvae, fish larvae, salps, and various classes of arthropods.

To investigate which species were responsible for the majority of the biomass at the site when overall biomass values in the water were high, samples were analysed taxonomically via microscopy. A subsample was prepared using a Stempel pipette, and organisms were identified and counted in a Bogorov tray using a microscope. Dominant members were identified to genus or species with assistance from trained plankton taxonomists at the CSIRO Plankton Ecology Laboratory (Eriksen et al., 2019).

Zooplankton size structure

A size distribution of the sample particles, known as a Normalised Biomass Size Spectra, was produced to analyse the size structure of the zooplankton community (Vandromme et al., 2012). Spherical biovolume was calculated from the size measurements obtained from ZooScan. Each particle was assigned to one of 50 logarithmic size categories based on its spherical biovolume. The sum of the spherical biovolume of the particles in each size class (mm3) was standardised by the fraction of sample scanned and the volume of water filtered (m3), and normalised by dividing this value by the width of the size class measured in biovolume (mm3). Both axes of the Normalised Biomass Size Spectra use a logarithmic scale.

Drivers of zooplankton biomass and manta numbers

To investigate potential drivers of zooplankton abundance and M. alfredi visits to Hanifaru Bay, we constructed generalised linear models (GLMs) using R (R Core Team, 2019). Separate analyses were conducted for two response variables: (i) Zooplankton biomass (mg m−3), with a Gamma error structure and log-link function; and (ii) Manta ray abundance (number of M. alfredi observed during zooplankton sampling), with a negative binomial error structure and log-link function (Poisson error structure was overdispersed). We visually inspected diagnostic plots to assess assumptions of homogeneity of variance and normality. Independent variables in both models were Tide (hours from high tide) and Behaviour (Feeding, Non-feeding and Absent). To account for the circular nature of Tide (∼12-hr cycle), the variable was transformed using a truncated Fourier series (a harmonic function of sines and cosines). This ensures that the cyclical nature of this variable is captured, while guaranteeing that the response values predicted at the extremes of the variable range are the same (i.e., the same prediction for Zooplankton biomass or Count at times of 0 and 24 h). For the Manta ray abundance model, the Behaviour variable was reduced to two categories—namely Feeding and Non-feeding, and Zooplankton biomass (mg m−3) was included as an independent variable. Models were plotted on the response scale using the package “visreg” in R (Breheny & Burchett, 2017).

Critical thresholds for feeding behaviour and strategy

We assessed whether there might be a critical threshold for M. alfredi feeding—i.e., a level above which the likelihood of feeding increases dramatically. We thus used a GLM with a binomial error structure to analyse manta ray behavioural response (Non-Feeding = 0, Feeding = 1) in relation to zooplankton biomass (mg m−3) as a predictor. The critical density threshold was taken as the zooplankton biomass at which the proportion of feeding was 0.5. A theoretical prey density threshold was plotted for comparison, based on findings by Armstrong et al. (2016). Their study assumed general morphometrics (average disc width of 3.5 m, mouth opening of 0.3 m, and mass of 100 kg) and swim speeds (2 knots when feeding) for M. alfredi. These assumptions are also applicable in the current study.

Feeding samples were categorised into either Solo feeding (Straight, Surface and Somersault) or Group feeding (Piggy-back and Chain) based on the most common strategy observed in the manta rays (Stevens, 2016). A GLM with a binomial error structure was used to analyse manta ray feeding strategy response (Solo = 0, Group = 1) in relation to zooplankton biomass (mg m−3) as a predictor. The critical density threshold was taken as the zooplankton biomass at which the proportion of Group feeding was 0.5.

Zooplankton community analysis

To determine how different the zooplankton communities were for the M. alfredi behaviours (Feeding or Non-Feeding), non-metric multidimensional scaling was used based on abundance counts of the different taxonomic groups from the Zooscan analysis. The Bray Curtis distance measure was used because it is unaffected by joint absences of taxonomic groups in samples. To account for abundance of certain taxa, data were transformed using a root transformation. To test for differences in community composition between M. alfredi behaviours (Feeding and Non-Feeding), we performed an adonis analysis, a multivariate analysis of variance. Both the adonis and non-metric multidimensional scaling were conducted using the “vegan” package in R (Oksanen et al., 2007).

Results

A total of 77 zooplankton samples were collected (Feeding = 33, Non-feeding = 22, and Absent = 22) over a period of nine days. Overall zooplankton biomass ranged between 0.7 and 643.1 mg m−3 (mean = 90.7, SD = 130.9). For manta ray behaviours, zooplankton biomass for Feeding samples ranged between 7.3 and 593.6 mg m−3, for Non-feeding samples between 1.1 and 175.6 mg m−3, and for Absent samples between 0.7- and 643.1 mg m−3.

GLM analyses showed that Zooplankton biomass in Hanifaru Bay was significantly related to Tide and Behaviour (Fig. 3). Zooplankton biomass was greatest just following high tide (t = −3.83, p = 0.0003, Fig. 3A), and M. alfredi was more commonly observed feeding when zooplankton biomass was higher (t = −2.83, p = 0.006, Fig. 3B).

Figure 3 Zooplankton biomass in Hanifaru Bay, Maldives.

Raw data including (A) Scatterplot of Zooplankton biomass and Tide (hours from high tide) and (B) boxplot of Zooplankton biomass and Manta ray behaviour (Feeding, Non-feeding and Absent). Model output of significant independent variables related to Zooplankton biomass, including (C) tide (hours from high tide) and Manta ray behaviour. Biomass is on the response scale, with 95% confidence intervals.

Manta ray behaviour was significantly related to zooplankton biomass (z = 3.08, p = 0.002), with a prey density threshold of 53.7 mg m−3 calculated for feeding M. alfredi (Fig. 4).

Figure 4 Critical prey density foraging threshold.

Logistic regression of reef manta ray Mobula alfredi behaviour (Feeding = 1, Non-feeding = 0) in relation to zooplankton biomass (mg m−3). The black dashed line represents the critical prey density threshold of zooplankton biomass required to trigger manta ray feeding from in situ sampling (53.7 mg m−3), and the red dashed line represents the theoretical prey density threshold calculated to meet the metabolic requirements of foraging M. alfredi (25.2 mg m−3; Armstrong et al., 2016).

Manta ray abundance was significantly related to Behaviour (z = −5.55, p = 0.000000003; Fig. 5), with more M. alfredi individuals present when they were feeding in Hanifaru Bay. Tide and Biomass did not significantly relate to manta ray abundance. Manta ray abundance ranged between 0 and 25 individuals.

Figure 5 Reef manta ray Mobula alfredi abundance in Hanifaru Bay, Maldives.

(A) Raw data boxplot of Manta ray abundance and Manta ray behaviour (Feeding and Non-feeding); and (B) Model output of significant variable related to greater manta ray numbers in Hanifaru Bay. Manta ray abundance is on the response scale, with 95% confidence intervals.

There was no significant difference in zooplankton biomass among different feeding strategies during the study (ANOVA: F = 1.02, df = 4,28, p = 0.41). In addition, there was no significant difference in zooplankton biomass, when samples were pooled into Solo feeding strategies (n = 23) and Group feeding strategies (n = 10, z = 0.98, p = 0.33). However, only groups were observed feeding when biomass concentrations exceeded 200 mg m−3.

Zooplankton community composition

There were differences in the zooplankton community composition between Feeding and Non-feeding samples when analysed using non-metric multidimensional scaling on the Zooscan taxonomic counts (Fig. 6A). The 95% confidence ellipses for Feeding and Non-feeding were not overlapping, implying that they were significantly different zooplankton community compositions, and this was confirmed by the adonis analysis (F = 9.42, df = 1,53, p = 0.001). Crustaceans (such as copepods) were more associated with Feeding samples, compared to gelatinous taxa (such as chaetognaths and eggs), which were more associated with Non-feeding samples.

Figure 6 Zooplankton composition and reef manta ray Mobula alfredi behaviour.

(A) Non-metric multidimensional scaling analysis of zooplankton community composition. Ellipses represent95% confidence intervals and broad taxonomic groups are labelled as per their association with manta ray behaviours. (B) Undinula vulgaris specimens (Credit: Julian Uribe-Palomino). Percentages of zooplankton community composition in Hanifaru Bay in relation to manta ray behaviour: (C) Feeding; and (D) Non-feeding. ‘Other’ comprises taxonomic groups that contributed less than 5% to the total community composition.

Calanoid copepods comprised 66.3% of Feeding samples compared to 46.7% of Non-feeding samples (Figs. 6C and 6D respectively). Chaetognaths were 5.3% of Feeding samples, and 11.9% of Non-feeding samples. Fish eggs were less than 2% of Feeding samples, and 13.0% of Non-feeding samples. Based on microscopy, Undinula vulgaris (juveniles and adults) was the dominant calanoid copepod species in both Feeding and Non-feeding samples (25.0% and 30.7% respectively, Fig. 6B).

Zooplankton size structure

Analysis of the size structure of zooplankton from Hanifaru Bay revealed that the biovolume of zooplankton increased in the majority of size categories when M. alfredi was feeding (Fig. 7). The biovolume of zooplankton was significantly higher across particle size categories during M. alfredi Feeding events than Non-feeding events (Mean total standardised biovolume: Feeding = 288.4 and Non-feeding = 172.1; t = −2.66, df = 51.38, p = 0.01). Feeding and Non-feeding samples had similar biovolumes of small and large particles, but Feeding had significantly more moderate-sized particles (from 10−1.2 to 100.5 mm3).

Figure 7 Zooplankton size structure analysis.

Normalised Biovolume Size Spectra of the zooplankton community when reef manta rays Mobula alfredi are Feeding (n = 33) and Non-feeding (n = 22). For the x-axis, each particle was assigned to one of 50 logarithmic size categories based on its biovolume (Size class). For the y-axis, the sum of the biovolume of the particles in each size class (mm3) was standardised by the fraction of sample scanned and the volume of water filtered (m3), and normalised by dividing this value by the width of the size class measured in biovolume (mm3; Normalised biovolume). Dashed lines represent standard error.

Discussion

Summary

Zooplankton concentrations influence the number of M. alfredi present and their observed behaviour in Hanifaru Bay. Rapid changes in zooplankton are observed across the tidal cycle, and M. alfredi feed when biomass reaches a critical density which is higher than predicted to meet their theoretical metabolic requirements. Mobula alfredi foraging occurs when the zooplankton community is dominated by calanoid copepods, and feeding is less likely when there are greater numbers of gelatinous taxa (such as chaetognaths or eggs). Taxonomic analysis reveals that the large-bodied copepod, Undinula vulgaris, dominates the zooplankton environment at Hanifaru Bay, suggesting oceanic incursions may play an important role in bringing zooplankton to this small reef inlet.

Tidal influence on zooplankton density and manta ray foraging

Manta rays feed when zooplankton biomass is high, which is typically observed on the high to ebbing tide at Hanifaru Bay. Oceanographic investigations in Hanifaru Bay suggest tidal currents draw zooplankton into the shallow reef systems of the atoll, where they become trapped inside due to a back-eddy mechanism created by the unique shape of the reef system and the combination of the lunar and monsoon currents (Hosegood pers comms). Tides are known to influence the distribution and abundance of zooplankton around island inlets in the Great Barrier Reef (Alldredge & Hamner, 1980), and have been shown to influence manta ray feeding behaviour at aggregation sites in Indonesia (Dewar et al., 2008), eastern Australia (Armstrong et al., 2016), and the Chagos Archipelago (Harris et al., 2021). Therefore, short-term in situ observations of zooplankton concentrations in relation to tidal cycles and manta ray behaviour can help inform when M. alfredi is likely to be observed in Hanifaru Bay.

Animal movements and productivity hotspots

Large planktivores seeking to exploit ephemeral food sources in surface waters are likely to respond to currents and water movements that concentrate zooplankton. Cetorhinus maximus forage along thermal fronts (Sims & Quayle, 1998), R. typus targets regions of upwelling (Ryan et al., 2017), and surface foraging in M. alfredi is often tidally driven (Dewar et al., 2008; Armstrong et al., 2016). In conjunction with responding to physical oceanographic cues, animals that seek patchily distributed prey sources are also likely to congregate in areas where they have previously encountered energetically rewarding prey abundances, resulting in larger numbers of animals in reliable foraging regions. For example, M. alfredi predictably switches to the down-current side of the atolls in the Maldives in response to monsoonal winds and primary productivity (Harris et al., 2020). Area-restricted search theory predicts that animals will remain localised in areas where they have a higher probability of encountering prey (Bailey et al., 2019), and this perhaps explains why some M. alfredi individuals remain in Hanifaru Bay when not feeding. This location also has two cleaning stations used by M. alfredi (Stevens, 2016), and it is hypothesised manta rays will frequent cleaning stations in close proximity to foraging opportunities (Armstrong et al., 2021). Manta rays in Palmyra Atoll used area-restricted searching when adjacent to ledges or channels with high plankton concentrations, but their movements were more random at larger spatial scales (Papastamatiou, De Salles & McCauley, 2012). Area-restricted searching has also been observed in two dolphin species (Tursiops truncatus and Delphinus delphis) in areas of high prey availability, and where they have had previous successful foraging experience, suggesting memory plays a role in their movement ecology (Bailey et al., 2019). For M. alfredi, the apparent preference for returning to the same cleaning stations over time (Armstrong et al., 2021), suggests they may form a cognitive map of shallow reef environments, and this is likely the case for known productivity hotspots as well.

High critical feeding threshold for manta rays at Hanifaru Bay

The critical prey density threshold for M. alfredi feeding in Hanifaru Bay (53.7 mg m−3) is more than four times higher than that in east Australia where M. alfredi feeds (11.2 mg m−3; Armstrong et al., 2016), and in east Africa where R. typus feeds (12.4 mg m−3; Rohner et al., 2015). It is also double the theoretical prey density threshold calculated to meet the metabolic requirements for M. alfredi (25.2 mg m−3; Armstrong et al., 2016), which may explain why this site hosts such a large feeding aggregation of this species. However, these large planktivorous elasmobranchs are assumed to feed in the mesopelagic layer (Couturier et al., 2013; Burgess et al., 2016), so an understanding of the prey densities available at these depths is required to gauge the relative importance of aggregations sites such as Hanifaru Bay in meeting these species’ daily energetic requirements. Sampling zooplankton at depth remains a logistical challenge for researchers, but with technological advances, such as satellite tags equipped with accelerometer data loggers, and unmanned video submersibles (Stewart et al., 2018), these inferences can be better investigated.

Manta ray feeding strategies

In the current study, plasticity in M. alfredi feeding strategies in response to changes in prey biomass in Hanifaru Bay is not supported. This contrasts with work previously conducted in Hanifaru Bay that found manta rays were significantly more likely to employ group feeding strategies as prey density increased (Stevens, 2016). The previous work was based on a qualitative visual index for prey density, with data obtained over a long time period (>5 years) and included aggregations upwards of 150 animals. However, we did see that when zooplankton biomass values were very high, over 200 mg m−3, that only Group feeding strategies were used, and no Solo feeding was seen. But in either scenario, it is uncertain whether the observations are due to true cooperative feeding strategies, or simply that coordinate movements reduce collisions with other manta rays (Stevens, 2016).

Zooplankton composition and size

Differences in the composition of the zooplankton community were observed between M. alfredi Feeding and Non-feeding events, and M. alfredi was observed feeding when the overall biovolume of zooplankton was greater. Calanoid copepods dominate the zooplankton community for manta rays at Hanifaru Bay during the time of the study. This is similar to findings for M. birostris at Isla de la Plata, Ecuador, where zooplankton samples collected during feeding and non-feeding events were dominated by calanoid copepods, but also cyclopoid copepods (Burgess, 2017). The diet of mobulids (M. japonica, M. thurstoni, M. tarapacana, and M. birostris) in the Philippines is more diverse, with euphausiid krill dominating, but also records from stomach content analysis of squid, fish and copepods (Rohner et al., 2017). Investigations into foraging aggregations of C. maximus off Plymouth in the UK have also revealed the dominance of copepods during feeding events, with threefold increases in this taxa reported (Sims, 1999). In contrast, the food environment at R. typus feeding aggregations appear to be more diverse, with R. typus feeding on sergestid shrimps, fish spawn, calanoid copepods, and small bait fish (Motta et al., 2010; Fox et al., 2013; Rohner et al., 2015). These findings suggest that the food environment for large planktivores feeding in predominately tropical or sub-tropical waters is varied, and is likely driven by pulses in productivity, rather than particular taxa.

Undinula vulgaris is the most prominent species observed in zooplankton samples from Hanifaru Bay. It is a key species in tropical areas due to its large size and tendency to swarm in high numbers, making it a good food resource for planktivorous fishes (Alvarez-Cadena, Suárez-Morales & Gasca, 1998). This species has been observed at numerous large planktivore feeding aggregation sites, including those visited by M. alfredi in eastern Australia (Couturier et al., 2013; Armstrong et al., 2016), R. typus in the Gulf of Tadjoura, Djibouti (Boldrocchi et al., 2018), both M. birostris and R. typus in the Gulf of California (Notarbartolo-di Sciara, 1988; Lavaniegos, Heckel & De Guevara, 2012), and both M. alfredi and R. typus in the Philippines (Canencia & Metillo, 2013; Yap-Dejeto et al., 2018). Undinula vulgaris is considered an indicator of the influence of neritic-oceanic waters in reef environments, and its local distribution can suggest oceanic water sources (Alvarez-Cadena, Suárez-Morales & Gasca, 1998). Further investigation into the ecology of U. vulgaris in tropical environments may aid our understanding of how vital swarms of this species are for supporting large tropical planktivores, and whether their distribution and abundance is likely to be impacted by a rapidly changing climate.

Caveats to zooplankton sampling

There were three main limitations in the sampling design of the current study. First, as boat engines are prohibited in the area, towing of the plankton was done by hand and this resulted in tow speeds slower than recommended (22 m vs 60 m per minute) for minimising capture-avoidance by larger zooplankton (Harris et al, 2000). However, the use of a net with a relatively large mouth area (50 cm diameter) would help mitigate this. Further, the use of slower towing speeds (Davies & Beckley, 2010) and a relatively large mesh size for tropical waters (200 µm; Eriksen et al., 2019) should reduce the pressure wave from the towed net, and thus reduce concerns about net avoidance by zooplankton. Second, zooplankton sampling was limited to surface waters. This could influence the findings, particularly in relation to investigating the zooplankton dynamics when manta rays are employing different feeding strategies. For example, somersault feeding and cyclone feeding strategies do not necessarily occur in surface waters. This issue could be overcome with the use of drop nets, vertical free-fall nets that sample on the way down with weighted rings for propulsion (Eriksen et al., 2019), to provide coverage throughout the water column. Lastly, it is likely our relatively short sampling duration failed to detect group feeding dynamics, and our results could suffer from small sample size. More work needs to be done to assess whether the presence of higher zooplankton biomass is positively correlated with group feeding events. In particular, higher sampling replication of the eight different feeding strategies is required to tease apart how these relate to zooplankton dynamics. One of the key issues for sampling design is that the magnitude and exact timing of the biggest feeding events each season is somewhat variable (Harris & Stevens, 2021). This highlights the need for studying multiple feeding events in a consistent way to distil the general pattern. To further examine the zooplankton and manta ray foraging dynamics at this site, we suggest a future sample size at least double the current study, and attempts should be made to sample larger feeding aggregations (50+ manta rays).

Conclusions

Identifying important foraging opportunities for vulnerable species such as manta rays remains a goal for implementing effective conservation strategies for the species. Here, we conducted the first analysis of the food environment for M. alfredi at Hanifaru Bay, and highlighted the importance of tidal regimes and high zooplankton density in driving M. alfredi aggregations at this site. Conducting high resolution investigations into the dietary basis of aggregations can help inform drivers of species movements and habitat preferences. This can be challenging in remote locations where resources are sparse, and where fieldwork is logistically difficult (i.e., hand-towing for zooplankton is seldom recorded in methods), which may provide an explanation as to why most feeding studies for marine megafauna have only superficially investigated zooplankton dynamics. This study failed to record the zooplankton dynamics during a mass feeding aggregation at this site, and so our findings are suggestive of what can be observed at this location, but they may not provide the whole picture. Longer term sampling, and more targeted methodologies that allow for sampling of zooplankton throughout the water column, will help elucidate what leads to mass feeding aggregations and the role of different feeding strategies for M. alfredi at this site. Nevertheless, the findings here have emphasised the importance of this site as a foraging ground for large aggregations of M. alfredi. To maintain this natural phenomenon, we suggest a number of management considerations. For example: (1) Implementing a code of conduct for tourism interactions with manta rays would help ensure human activities do not interfere with manta ray foraging activity; and (2) To preserve the zooplankton community, the oceanographic conditions of the region should not be altered (i.e., no dredging or alterations to natural sand movements). Climate change also poses an unknown risk to this aggregation site, as our findings suggest the zooplankton originate elsewhere and could be altered by predicted temperature shifts. Here, we have determined the importance of zooplankton dynamics in driving aggregative behaviour of M. alfredi at the large aggregation site in Hanifaru Bay, and this information can help inform management of this ecologically and economically important marine protected area and core zone within a UNESCO Biosphere Reserve.

Supplemental Information

Supplemental Information 1 Zooplankton biomass and reef manta ray survey data from Hanifaru Bay, Maldives, 2017

Click here for additional data file.

We would like to acknowledge the logistic and field assistance from the team at the Manta Trust, specifically Niv Froman and Tam Sawers. Special thanks to Stephanie Venables for assistance in the field. Thanks to the Plankton Ecology Lab in Brisbane for their expert assistance, namely Frank Coman. We would like to thank Simon Pierce and David Sims for providing valuable feedback on the manuscript.

Additional Information and Declarations

Competing Interests

Author Contributions

Field Study Permissions

Data Availability

Guy M.W. Stevens and Annie Murray are employed by the Manta Trust. The authors declare that there are no other competing interests.

Asia O. Armstrong conceived and designed the experiments, performed the experiments, analyzed the data, prepared figures and/or tables, authored or reviewed drafts of the paper, and approved the final draft.

Guy M.W. Stevens conceived and designed the experiments, performed the experiments, authored or reviewed drafts of the paper, and approved the final draft.

Kathy A. Townsend conceived and designed the experiments, authored or reviewed drafts of the paper, and approved the final draft.

Annie Murray, Amelia J. Armstrong, Julian Uribe-Palomino and Phil Hosegood performed the experiments, authored or reviewed drafts of the paper, and approved the final draft.

Michael B. Bennett and Christine L. Dudgeon analyzed the data, authored or reviewed drafts of the paper, and approved the final draft.

Anthony J. Richardson conceived and designed the experiments, analyzed the data, prepared figures and/or tables, authored or reviewed drafts of the paper, and approved the final draft.

The following information was supplied relating to field study approvals (i.e., approving body and any reference numbers):

Approval for the research was received from the Maldives Ministry of Fisheries [Permit No. (OTHR)30-D/PRIV/2017/280] and the Maldives Ministry of Environment [Permit Nos. EPA/2017/RP-01 & EPA/2016/PSR-M02].

The following information was supplied regarding data availability:

Zooplankton data (including biomass, taxonomy and zooscan data) is available at the UQ eSpace public repository:

Armstrong, Asia(2021). 2017 Maldives Zooplankton. The University of Queensland. Data Collection. https://doi.org/10.14264/98ddbde.

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
