# Peer review of "Reef manta rays forage on tidally driven, high density zooplankton patches in Hanifaru Bay, Maldives"

_PeerJ, doi:10.7717/peerj.11992_

## Round 0.1 · original submission · Minor Revisions

Please revise your manuscript to address all concerns of the reviewers.

·

Basic reporting

As below.

Experimental design

As below.

Validity of the findings

As below.

Additional comments

I very much enjoyed reading this study on what is clearly an incredible natural event. It's an excellent addition to the general literature on manta ray ecology and behaviour, as well as improving understanding of the drivers of this economically important feeding aggregation. The study is well-written and clearly presented.

As someone coming to this 'fresh', without any personal knowledge of Hanifaru, I do have a few general suggestions on additional context for the broader readership.

While this was a short-term study, it'd be helpful to have more background on why this specific timing was chosen. Do these conditions only take place on a seasonal basis at a specific moon phase? That would be useful ancillary information for interpreting manta residency and movements with regards to the site. Guy's thesis is cited for more information, but it would still be good to include summary information within this study.

Manta metabolic requirements are an important discussion point (and in Fig. 4), but the assumptions underlying the 2016 paper's calculations do not appear to have been critically re-examined here for this new geographical area and study population. For instance, I note that the 2016 estimate included assumptions on manta sizes, body mass, and swim speeds while feeding. I realise this wasn't a key part of this study itself, but it would be worth at least mentioning this in the Discussion, or preferably adding a Methods subsection that demonstrates the applicability of this value to Hanifaru mantas.

I like the use of hand-towed nets! However, it'd be good to include a quick mention somewhere of whether this may affect the results in any way – how fast are Undinula? Can they evade a small net? Does their behaviour influence manta feeding tactics at different densities? The aims don't explain 'why' you were anticipating why a change in manta tactics might occur.

The Results summary is good, but it would be useful to include more specific values, potentially on Fig. 3 and in the text: what was the lowest biomass at which mantas were observed feeding? The highest biomass at which mantas were absent, or present but not feeding? It's hard to pull this from Fig. 4. Similarly, I don't see non-response-scale data on manta abundance. It's all in the Supplemental data of course, but adding these data to the main MS would be helpful.

The importance of these results to the management of the site is mentioned in the Conclusions (L416–17). My understanding is that Hanifaru is an important area for tourism, and that access and activity restrictions are in place. It would be good to be more explicit with recommendations on how this new understanding of plankton and manta ecology could inform management and tourism activities going forward.

Specific comments:

L37 – I'd reverse this to say that high zooplankton biomass is a significant predictor of feeding events (i.e. L257–258).

L46 – This sentence could be broken up for clarity.

L94 - Prebble et al. (2016) is a conference abstract. Better to use a published work here.

L111 – Given that Hanifaru is an important tourism area, it'd be good to include more specifics on both the situation and aims in terms of local conservation and management.

Figure 3: Surely manta behaviour is correlated with zooplankton biomass, not a 'predictor' in real terms?

Figure 3b: This seems like a very confusing way to present these data – with the biomass on the vertical axis it looks like 'real' data, i.e. Feeding at ~100–~310 mg.m3, but this is actually showing the confidence intervals? A figure that accurately summarises the raw data would be preferable for interpretation. I have the same basic comment about Fig. 5.

I hope this is generally helpful.
Simon Pierce.

·

Basic reporting

General comments
This study analyses zooplankton samples taken in the presence and absence of feeding/non-feeding and solitary/grouped reef manta rays (Manta alfredi) in the Maldives to achieve the stated aim to improve understanding of the feeding dynamics of the species. Zooplankton samples were taken in an semi-enclosed bay over a week or so when mantas were present and conducting various behaviours in addition to when they were absent. The zooplankton biomass, size structure and community composition were determined to establish critical prey density thresholds, whether prey density influences feeding strategy, and, lastly, whether behaviour changes with zooplankton characteristics.

This is a well written paper that presents results of value to the field. The design of the study is good, not least in trying to mitigate the logistic difficulties of working on wild foraging ecology of large planktivores. The methods used are appropriate to this high degree of difficulty. The zooplankton analysis methods in particular are comprehensive and open up new questions and potential comparisons with other plantkivores that progresses the field. I recommend the paper for publication after some minor additions and points of clarification.

Specific comments (by line number)
L38. Abstract. Need to state that the threshold is expressed in mg DW m-3.
L59. Need to clarify that it is “often difficult for researchers..”.
L60-61. Need a review reference or two to substantiate this statement given it refers to planktivores generally.
L75-76. Need to reference other studies on planktivores where this has been done to support your view that this is a valid approach.
L143. What was the tow speed? Even an estimate would be useful because it may be useful to readers to be able to assess the possibility of zooplankton escaping small-diameter nets at slow tow speeds. Check the net diameter is given too.
L143-145. Need to give the numbers of replicates taken per feeding/non-feeding event and for each behaviour type if this was possible, or state why it wasn’t and what the limitations might be as a consequence (can add to Discussion of limitations). Important for assessing the representativeness of a sample assigned to a feeding/non-feeding event or behaviour type.
L159. The source of tide data is given as “..from a local government representative”. However this gives no opportunity for precise replication. Need to give the government department name at the least.
L181. Why was a cactus spine used? Is this important in some way to the analysis? Please clarify or remove.
L267-271. Need to discuss or at the very least acknowledge the sampling limitations. For example, that zooplankton biomass was not different for feeding strategies or solo or group behaviour may be down to too few samples and lack of replicates for particular behaviour events. Add sentences to Discussion to acknowledge potention limitations.
The zooplankton size structure and community composition results are strong elements of the paper. These are not easy to quantify but this is done very well; well done.
L359. Usually this net energy gain/loss relates to “daily energetic requirements”.
L375. To help future research, it would be useful to suggest what you think would be a suitable sample size to test the hypothesis.
L379-396. The Discussion would benefit from broadening to include a few sentences to compare what was found with other large planktivores for which copepods have been identified as being important identifiers of feeding conditions and for which critical swithing thresholds have been determined. For example, did those swarms also comprise of larger numbers of copepods with greater body sizes in the mid-range of the size distribution? Could there be general rules across planktivores?
L.400 onwards. There is no critical discussion of the study’s limitations. This should be added with a few sentences, particularly on sample size and replicates. It is mentioned that hand-towing of a plankton net is not mentioned in the records, but it isn’t discussed why this might be? It is possible hand-towing isn’t used because too many large zooplankton escape as tow speeds are slow. So it would be useful to provide your view on the limitations and why in your situation it was a valid technique for sites not allowing engines. Also, it may also help other researchers to know how in your experience vertical samples could be taken at such sites.
Fig. 1. Adding to the map a schematic of the main water movement directions/currents occurring in the study site (Hanifaru Bay) would help readers to understand the complexity of the site.
Fig. 7. Explain in the legend the term biovolume and its units, that is, how specifically it is different from body volume.

Experimental design

See previous section

Validity of the findings

See previous section

---

## Round 0.2 · accepted · Accept

Thank you very much for your conscientious attention to the concerns of the reviewers, and congratulations on a fine paper!